# The Correlation between Convenience Stores' Distribution and Urban Spatial Function: Taking the FamilyMart Stores in Shanghai as an Example

**Shuling Liao, Kun Liu \*, Ying Yang 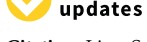 and Yong Liu**

Shanghai Academy of Fine Arts, Shanghai University, Shanghai 200444, China; surin123@126.com (S.L.); yangyangklwx@126.com (Y.Y.); liuy533@shu.edu.cn (Y.L.)
\* Correspondence: liukun_arch@shu.edu.cn

**Abstract:** This manuscript mainly focuses on the correlation between the spatial distribution of convenience stores and the distribution of urban spatial functions. We took 1217 FamilyMart convenience stores within the Shanghai outer ring area as the research objects and used the calculation method of kernel density estimation. The results show that the distribution of Shanghai FamilyMart stores is centered on the three central urban areas of Huangpu District, Jing'an District, and Xuhui District; the high-density areas of convenience stores partially overlap with the city's main center, and some do not. The research shows that in areas with a high distribution density of convenience stores, the establishment of the convenience stores has a strong correlation with urban spatial functions, and these continence stores have formalized a network deeply integrated with the community. We believe that, in the future, with the development of small retail businesses, these can be transformed into public activity spaces through space transformation and design, which promotes the sustainable development of community interaction and traditional retail business, stimulating the sustainability of community vitality. The results of this study show that this concept has a feasible spatial basis, and the research method can also be applied to the study of other small retail commercial spaces.

**Keywords:** small retail business; urban spatial distribution; community interaction; Shanghai; FamilyMart

## 1. Introduction

With the rapid rise of new retail in China, a new retail model, with a deep integration of online service, offline experience, and modern logistics, has emerged. More and more Internet companies have invested in the traditional supermarket industry and guided them to realize Internet transformation, such as Alibaba's Hema Xiansheng, JD.com's stake in Yonghui Supermarket, Amazon's acquisition of Whole Foods Market, etc. At the same time, under the background of current online retail encountering rising bottlenecks, the difficult development of brick-and-mortar retail, and the emergence of the urban communication crisis, traditional community retail is also exploring a direction for transformation. We found that in the current process of urban community space renewal, as the daily living space of residents, community public space still has problems, such as spatial simplification, excessive typification, and a low sense of belonging for residents. It is manifested in the lack of flow of people in the community public space and the low utilization rate, which cannot meet the diverse needs of community residents.

Through on-the-spot research, we found that small retail businesses in the community are gathering places for community residents. Whether they are in-office communities or residential communities, they all have the characteristics of high traffic and high frequency of use. Residents have strong communication needs in small community retail businesses. For example, community coffee shops, bookstores, and retail stores will attract large numbers of people. People stop here to rest and chat. Therefore, we believe that small

retail commercial space has the possibility to become a community communication space. This can not only help the traditional retail and commercial space to effectively transform and achieve sustainable development, but also stimulate the sustainable vitality of the community public space. At present, there are few existing studies on the combination of community retail commercial space and community interaction space. Therefore, this research has a certain timeliness and pertinence.

We have selected a representative small retail business, FamilyMart, which is the oldest small retail business with the largest volume and widest distribution in Shanghai. First, we discuss its overall spatial distribution characteristics. Additionally, we further compare its spatial distribution with the urban spatial function distribution to verify whether the two have formed a strong coupling relationship. Conditioned to provide a space foundation for the occurrence of community public communication activities, the organization can undertake more community communication functions in the future and realize the sustainable development of traditional retail commercial space.

## 2. Related Studies

At present, commercial space research in China is mainly divided into macroscopic research and mesoscopic research. Based on location theory and business management theory, the influencing factors and factors of commercial location selection are discussed from the perspectives of society, politics, economy, history, culture, consumer behavior, formation mechanisms, etc. In the 1980s, the commercial retail space was mainly studied from a macro level, and qualitative analyses of the city's commercial outlets were carried out [1]. At the beginning of the 21st century, the research vision gradually extended from the macro to the meso level, and the location selection of commercial space became the mainstream research object [2]. From the research on convenience stores in Japan by Jiang Kaikai and others, the distribution of stores in various administrative districts of Tokyo is relatively balanced, and there are no obvious differences due to economic development level, population density, or other reasons [3]. At this stage, scholars worldwide have performed fruitful research on commercial space, and the research has gradually focused on the spatial structure and location selection of different commercial formats. In recent years, due to the continuous developments in mass data collection technology and Big Data analysis tools such as GIS technology, the research trend has gradually turned to the quantitative and visual analysis of selecting commercial space locations.

Research on commercial space in China started relatively late. Zhu Feng et al. took 160 large-scale department store retail commercial facilities as the research objects and used GIS technology to analyze the spatial layout characteristics, influencing factors, and the commercial center system of Shanghai Pudong New Area [4]. Taking the 7–11 convenience store as an example, Zhou Qianjun and others used methods such as questionnaire surveys and field observations to study the spatial layout and residents' utilization characteristics of convenience stores in the Beijing urban area [5]. Xiao Chen et al. took Nanjing Suguo Supermarket as the research object and used GIS analysis technology and measurement methods to analyze the spatial distribution characteristics of chain supermarkets in large cities and the influencing factors of the location selection of different formats [6]. Taking four convenience stores in Shanghai as the research objects, Li Xue et al. analyzed the spatial distribution and influencing factors of convenience stores by using POI points, spatial analysis methods and spatial regression models [7]. Taking Shanghai FamilyMart as an example, Hao Kuo et al. used spatial analysis and quantitative statistical methods to analyze the main factors affecting the distribution of FamilyMart [8]. Taking Shanghai FamilyMart convenience store as an example, Guo Weipeng et al. objectively analyzed the current energy consumption level and characteristics of convenience stores through survey collection and the statistical analysis of energy consumption data [9].

In recent years, international research directions have gradually diversified. For example, in the article entitled "Regional spatial structure and retail amenities in the Netherlands", Martijn et al. studied the interaction between retail business and the overall

spatial structure of the region [10]; in the article entitled "A GIS-Based Approach for Catchment Area Analysis of Convenience Store" by Dyah Lestari Widaningrum, the author mainly used GIS technology to study the relationship between the spatial distribution of convenience stores, population characteristics and social economy [11]; in the article entitled "Retail store formats, competition and shopper behavior: A Systematic review" by André Bonfrer et al., the authors mainly studied the driving factors behind consumers' choices of convenience stores and their reactions to cross-store marketing activities [12]. In the article entitled "Retail concentration: a comparison of spatial convenience in shopping strips and shopping centers", Vaughan Reimers et al. proposed the concept of "retail concentration" to study the convenience provided by shopping malls and shopping street [13]; in the paper entitled "Bi-level Optimization Model of O2O Fresh Product Experience Store Location Considering Service Quality", Wang Xuping et al. studied the suitable location layout of O2O fresh product experience stores [14].

In recent years, attention to new retail business has gradually increased, and a certain research foundation has been established; however, most of the research still focuses on the layout structure and influencing factors of comprehensive commercial space at the macro and meso levels. Less attention is paid to retail formats, and empirical research on the spatial agglomeration characteristics of different types of retail businesses based on POI data is insufficient; it is even rarer to conduct correlation research combining the spatial distribution of convenience stores with specialized formats and the distribution of urban spatial functions.

In view of this, based on the POI data of Shanghai FamilyMart convenience stores, taking 1217 FamilyMart convenience stores in the outer ring of Shanghai as the research objects, and using the calculation method of kernel density estimation, the distribution characteristics of convenience stores were obtained. Then, high-density areas where convenience stores were more concentrated were selected, and correlation research on the distribution characteristics of different types of convenience stores and urban spatial functions was conducted.

## 3. Methods

### 3.1. Data Sources

Convenience stores are retail businesses that mainly operate daily necessities and serve residential areas. The area ranges from 50 to 200 square meters. They have the characteristics of long business hours, small-scale operations, and convenience. According to the definition of a convenience store in the "Convenience Store Handbook" compiled by Japanese small- and medium-sized enterprises in 1972: "In a small business district, a small store that sells daily necessities for a long time in a small business district" [15]. Additionally, the sales share of fresh food is less than 30%, the business hours are more than 16 h, and the business area is less than 200 square meters [5].

The main consumer groups of Japanese convenience stores are students and office workers under the age of 30, accounting for about 80% of the total number of customers. These groups have strong spending power, pursue fresh and fashionable products, and will readily accept new things; therefore, convenience stores in Japan are mostly located in large urban communities, schools, or business districts. According to the "China Convenience Store Development Report 2020", the current convenience store market is not yet saturated, and there is room for exploration. The future development trend of the convenience store industry is as follows: the format of convenience stores will maintain rapid growth, whereas the concentration of convenience stores will increase, and cooperation and integration will be a major trend. There is considerable space for expansion. In the face of omni-channel market competition, the edges of the business format are blurred, the community of convenience stores will become an important scene, and community group buying will become more popular [16].

Shanghai FamilyMart originated as Japan FamilyMart and entered the Shanghai retail market in July 2004. As of 30 April 2021, FamilyMart has a total of 1735 stores

in Shanghai, as shown in Figure 1, making it the largest and most widely distributed convenience store brand in Shanghai. Among them, there are 1217 stores in the outer ring of Shanghai, accounting for 70.14% of the total. Therefore, the research object was set as the 1217 FamilyMart convenience stores in the outer ring of Shanghai.

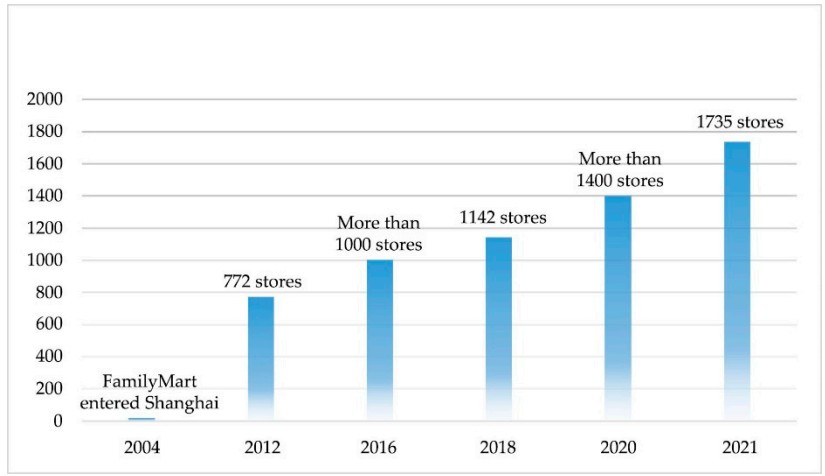

**Figure 1.** Number of FamilyMart stores in Shanghai over the years.

The main sources of data in this study were the official website of FamilyMart, the geographic coordinate data from Baidu Maps of FamilyMart in 2021, and the point of interest (POI) data of FamilyMart. The crawling and processing process was as follows: first, the coordinates were obtained from the Baidu Map system to obtain geographic location information data; secondly, Baidu panoramic map comparison and manual screening were performed to delete unqualified and duplicate data; next, ArcMap software was utilized to convert the geographic coordinate system to obtain the total effective POI data of the 1735 Shanghai FamilyMart convenience stores (data occurred in April 2021); thus, in the boundary of the Shanghai outer ring, 1217 datasets of FamilyMart convenience stores in the Shanghai outer ring were counted, as shown in Figure 2.

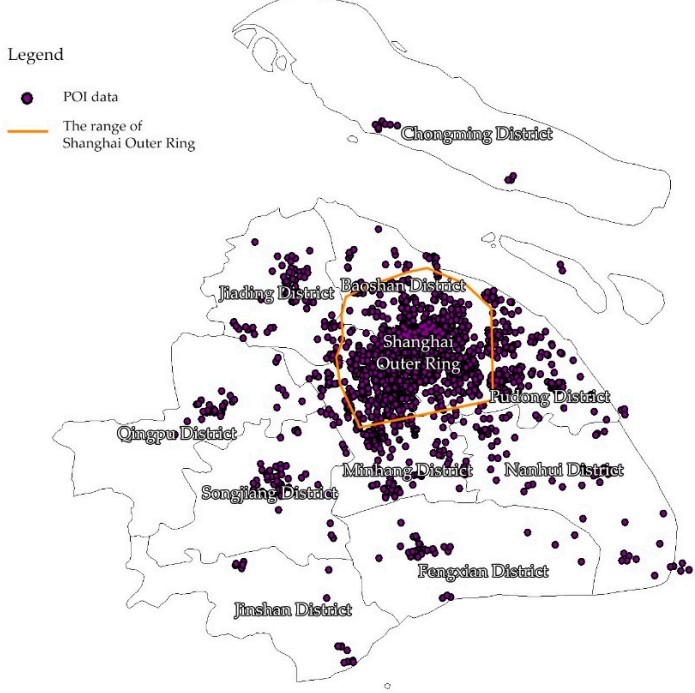

**Figure 2.** POI data of Shanghai FamilyMart stores.

### 3.2. Qualitative and Quantitative Approach

This study used ArcGIS software to calculate the kernel density of the data, i.e., counting the number of convenience stores per million square meters, the number of different types of convenience stores in high-density areas, and the ratio of urban spatial functions in high-density areas. This is a quantitative analysis method for the statistical analysis of various types of data. Describing the status quo of the distribution of FamilyMart convenience stores and analyzing the factors affecting the distribution of convenience stores formed part of the qualitative analysis. Through qualitative, comparative analysis, the spatial distribution of convenience stores and the distribution of urban spatial functions were assessed. The purpose was to analyze the correlation between the spatial distribution characteristics of convenience stores and the distribution of urban spatial functions. The combination of qualitative and quantitative research methods is helpful for the more accurate and deeper relationship between the spatial distribution of convenience stores and the distribution of urban spatial functions.

#### 3.2.1. Kernel Densitometric Analysis

In this study, kernel density analysis was used to calculate the density distribution of convenience stores. Kernel density analysis is mainly used to calculate the unit density of measured values of point and line elements in surrounding neighborhoods, which can be used to directly reflect the distribution of the measured value in a continuous area. In this analysis, points in different areas were given different weights; the points closer to a specific center were given greater weights. Through the calculation of each element point in the area, the kernel density distribution of the entire area was obtained. The higher the kernel density, the denser the distribution of feature points, and vice versa, the more scattered the distribution of feature points [17]. In this study, the factor points of FamilyMart were projected onto the Shanghai administrative base map, the sum density values of different points were calculated, and the spatial distribution of FamilyMart in the outer ring of Shanghai was investigated. The formula for calculating the kernel density is as follows:

$$f(x) = \sum_{i=1}^{n} \frac{1}{nh} k\left(\frac{d_{is}}{h}\right)$$

Here, $f(x)$ is the kernel density function, "$h$" is the distance attenuation threshold (i.e., bandwidth), "$n$" is the number of elements within the search distance, "$k$" is the spatial weight function, and "$d_{is}$" is the distance from the center point "$I$" to the points ($d_{is} < h$) [18].

#### 3.2.2. Comparative Analysis

Through calculation of the core density, the high-density area of convenience stores and the urban spatial structure were compared and analyzed to judge the relationship between the high-density area of convenience stores and the main city center. Then, the distribution locations of convenience stores in high-density areas were divided into three categories—commercial ground floor residential buildings, office buildings, and public buildings—and the number and proportion of different distribution locations of convenience stores in five high-density areas were counted. At the same time, the urban space functions of high-density areas were divided into three categories: residential space, office space, and public space. The building area of each function was counted, and the proportion of each function was calculated. Then, the proportions of each distribution location of convenience stores in high-density areas and the proportions of various functions in high-density areas were compared and analyzed to verify the coupling relationships between the spatial distribution of convenience stores and urban spatial functions, as shown in Figure 3. The formula for calculating the total building area of each spatial function of the city is as follows:

$$F = S \times H$$

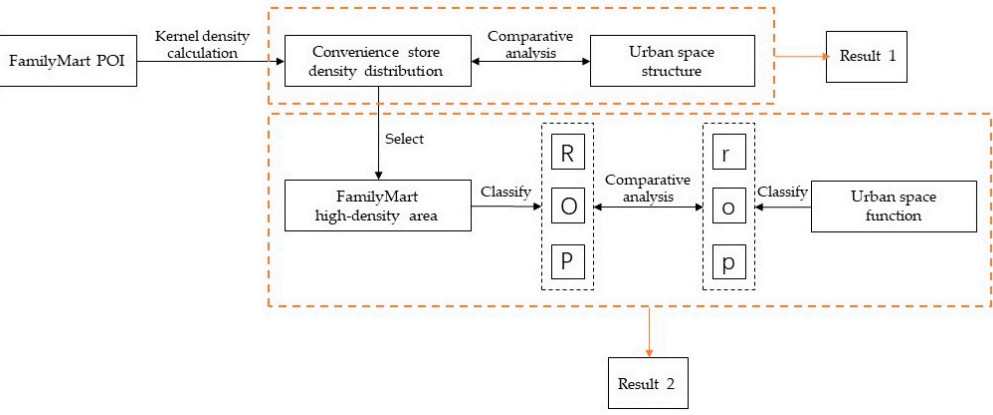

**Figure 3.** Research ideas and methods.

Here, "*F*" is the total construction area (m$^2$), "*S*" is the floor area of the building (m$^2$), and "*H*" is the height of the building.

## 4. Results

### 4.1. Overall Trend

Using the ArcGIS software to calculate the kernel density of the POI data of FamilyMart, we first analyzed the FamilyMart stores in Shanghai and obtained the kernel density distribution map, as shown in Figure 4. It was found that convenience stores were mainly concentrated in the outer ring of Shanghai. Then, the kernel density analyses of convenience stores in the outer ring and inner ring of Shanghai were carried out.

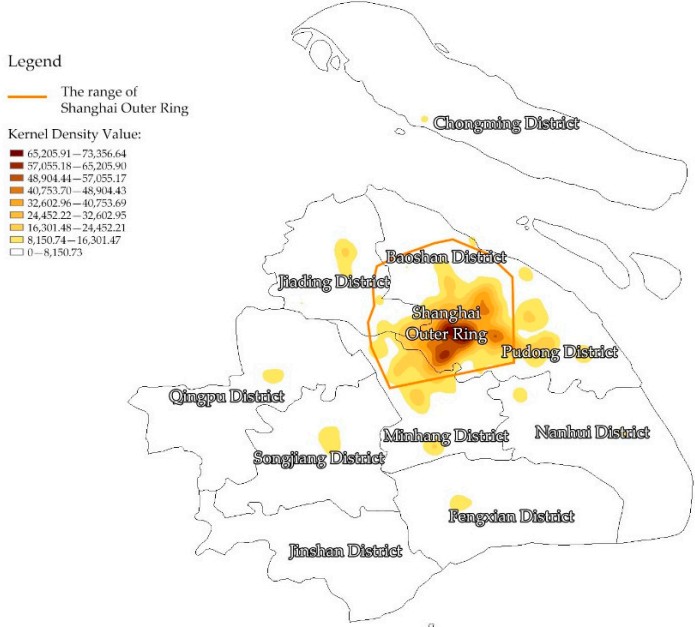

**Figure 4.** Kernel density analysis of Shanghai FamilyMart stores.

As shown in Figure 5, it can be seen that the high-density areas were mainly concentrated in Huangpu District, Jing'an District, and Xuhui District, followed by Hongkou District, Yangpu District, Pudong New District, Putuo District, Changning District, etc. The nuclear density level gradually decreased from the central area to the surrounding areas. Table 1 lists statistics on the number of FamilyMart convenience stores per square kilometer in each district within the outer ring of Shanghai. The numbers of stores per square kilometer in Huangpu District, Jing'an District, and Xuhui District were the highest in the outer ring of Shanghai, which was consistent with the results of the nuclear density analysis.

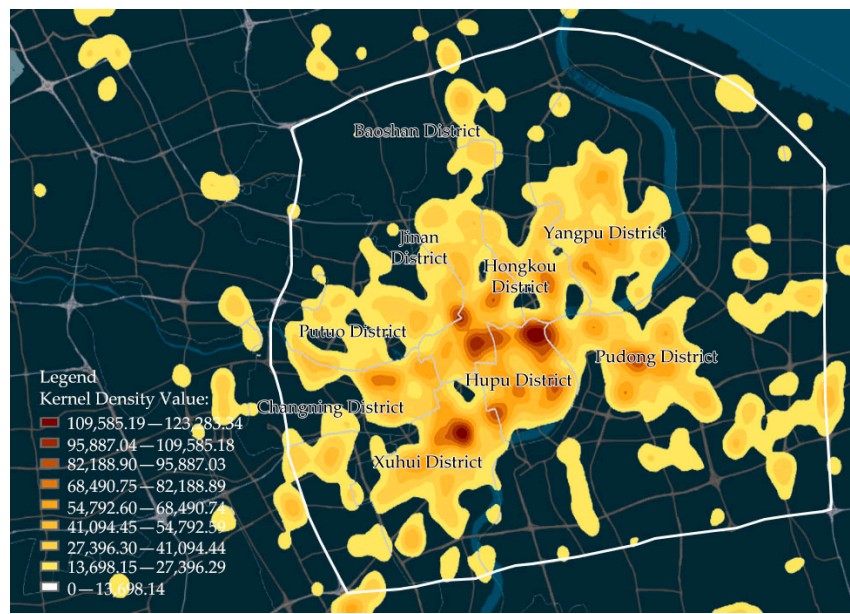

**Figure 5.** Kernel density analysis of FamilyMart stores in the Shanghai outer ring.

**Table 1.** Number of FamilyMart stores per square kilometer in the Shanghai outer ring.

| District | Number of Convenience Stores | Percentage | Area in the Shanghai Outer Ring (km$^2$) | Number of FamilyMart Stores Per Square Kilometer [1] |
|---|---|---|---|---|
| Huangpu District | 99 | 8.1% | 20.52 | 4.8 |
| Jinan District | 126 | 10.4% | 37.37 | 3.4 |
| Xuhui District | 155 | 12.7% | 49.74 | 3.1 |
| Changning District | 98 | 8.1% | 36.23 | 2.7 |
| Hongkou District | 62 | 5.1% | 23.45 | 2.6 |
| Putuo District | 112 | 9.2% | 55.15 | 2.0 |
| Minhang District | 89 | 7.3% | 45.64 | 2.0 |
| Yangpu District | 113 | 9.3% | 60.61 | 1.9 |
| Jiading District | 18 | 1.5% | 14.93 | 1.2 |
| Pudong District | 258 | 22.2% | 261.26 | 1.0 |
| Baoshan District | 87 | 7.2% | 174.08 | 0.5 |
| Total | 1217 | 100.0% | | |

[1] Number of FamilyMart stores per square kilometer = number of convenience stores/area in Shanghai outer ring.

The distribution locations of FamilyMart convenience stores in the outer ring of Shanghai were divided into three categories—commercial ground floor residential buildings, office buildings, and public buildings—as shown in Figure 6. There were 718 street-facing stores located on the ground floor of residential buildings, accounting for 59.00% of the total number; there were 278 stores located in office buildings, accounting for 22.84% of the total; 221 stores were in public buildings such as subway stations and commercial plazas, accounting for 18.16% of the total, as shown in Table 2. Therefore, FamilyMart convenience stores in the outer ring were mainly distributed in street-facing stores on the ground floors of residential buildings.

**Table 2.** Distribution locations of FamilyMart stores in the Shanghai outer ring.

| Store Location | | Quantity | Percentage |
|---|---|---|---|
| Commercial ground floor residential buildings | | 718 | 59.0% |
| Office building | | 278 | 22.8% |
| Public building | Subway station | 133 | 10.9% |
| | Emporium | 88 | 7.2% |

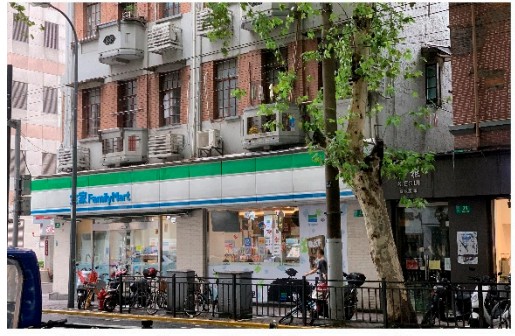
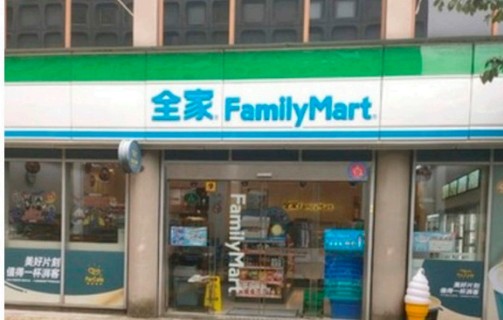

(a) Stores in residential, ground floor
(b) Stores in office building

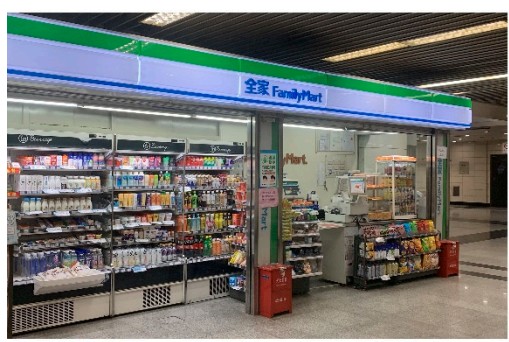
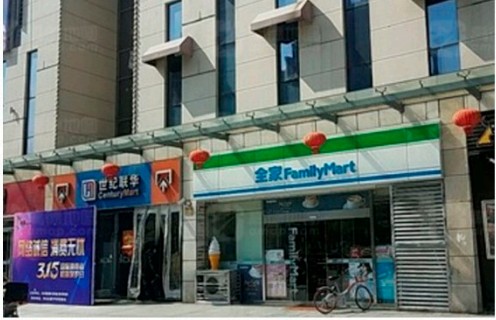

(c) Stores in subway station(public building)
(d) Stores in emporium(public building)

**Figure 6.** Example of Shanghai FamilyMart locations.

### 4.2. Relationship with the City's Main Center

By analyzing the high-density areas in Huangpu District, Jing'an District, and Xuhui District, we found that the Nanjing East Road–The Bund area in Huangpu District, the Nanjing West Road–Xinzha Road area in Jing'an District, the Xujiahui–Xujiahui Garden area in Xuhui District, the Jiangning Road area in Putuo District, and the Yanggao Road area in Pudong New Area were the most densely distributed areas, i.e., the "hot spots" of FamilyMart convenience stores in the outer ring, which can be seen in Figure 6. "A" stands for the Nanjing East Road–The Bund area, "B" stands for the Nanjing West Road–Xinzha Road area, "C" stands for the Xujiahui–Xujiahui Garden area, "D" stands for the Jiangning Road area, and "E" stands for the Yanggao Road area. According to the "Shanghai Urban Master Plan (2017–2035)", the city's main center includes 16 areas, as shown by the blue dots in Figure 7.

The high-density areas A, B, and C of convenience stores in the outer ring of Shanghai overlapped with the main city center, whereas areas D and E did not overlap with the main city center. Through quantitative analysis, FamilyMart convenience stores near the D area were mainly distributed on the ground floor of residential buildings and office buildings, each accounting for approximately 50%. In area E, 50% of the FamilyMart stores were located in office buildings, and 40% were located on the ground floor of residential buildings facing the street (Table 3).

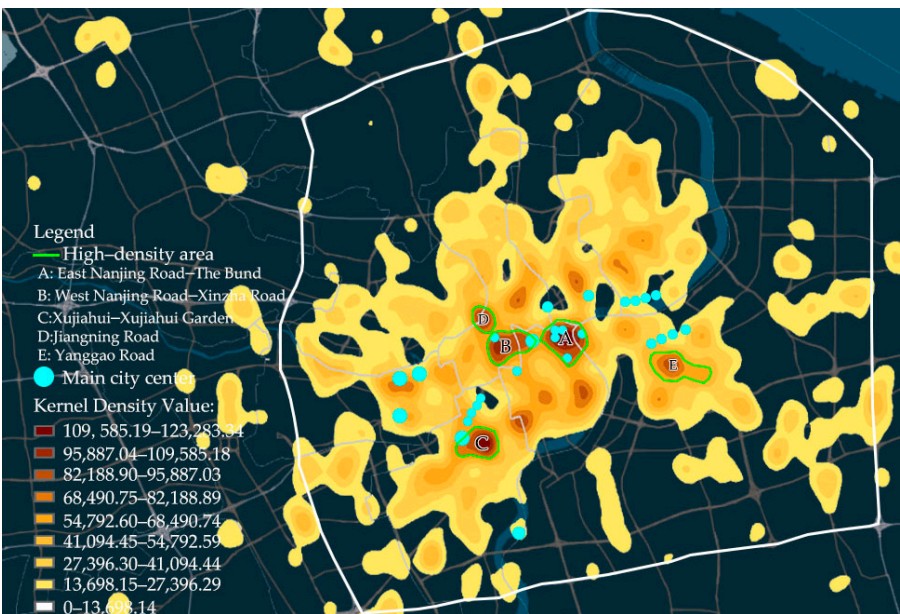

**Figure 7.** High-density areas of FamilyMart stores in the Shanghai outer ring.

**Table 3.** Distribution of FamilyMart stores in high-density areas of the Shanghai outer ring.

| High-Density Area | Store Location | Quantity | Percentage | Functional Classification of Cities | Occupied Area ($m^2 \times 10^5$) | Percentage | Gross Floor Area ($m^2 \times 10^5$) | Percentage | Number of Stores per Square Kilometer |
|---|---|---|---|---|---|---|---|---|---|
| A [1] | R [6] | 14 | 40.0% | r [9] | 4.2 | 41.2% | 16.6 | 22.1% | 8.4 |
| | O [7] | 15 | 42.9% | o [10] | 3.1 | 30.6% | 38.3 | 50.9% | 3.9 |
| | P [8] | 6 | 17.1% | p [11] | 2.8 | 28.1% | 20.3 | 27.0% | 3.0 |
| | Total | 35 | 100.0% | total | 10.1 | 100.0% | 75.2 | 100.0% | 4.7 |
| B [2] | R | 18 | 62.1% | r | 6.1 | 66.3% | 38.7 | 51.3% | 4.7 |
| | O | 8 | 27.6% | o | 1.4 | 15.7% | 23.2 | 30.8% | 3.4 |
| | P | 3 | 10.3% | p | 1.7 | 18.0% | 13.5 | 17.9% | 2.2 |
| | Total | 29 | 100.0% | total | 9.24 | 100.0% | 75.4 | 100.0% | 3.8 |
| C [3] | R | 8 | 42.1% | r | 2.7 | 50.3% | 27.9 | 55.6% | 2.9 |
| | O | 5 | 26.3% | o | 1.4 | 26.5% | 15.4 | 30.7% | 3.3 |
| | P | 6 | 31.6% | p | 1.2 | 23.2% | 6.9 | 13.8% | 8.7 |
| | Total | 19 | 100.0% | total | 5.4 | 100.0% | 50.2 | 100.0% | 3.8 |
| D [4] | R | 5 | 50.0% | r | 2.8 | 73.1% | 23.7 | 68.4% | 2.1 |
| | O | 5 | 50.0% | o | 0.8 | 19.7% | 8.9 | 25.9% | 5.6 |
| | P | 0 | 0.0% | p | 0.3 | 7.3% | 2.0 | 5.7% | 0.0 |
| | Total | 10 | 100.0% | total | 3.9 | 100.0% | 34.6 | 100.0% | 2.9 |
| E [5] | R | 8 | 40.0% | r | 2.0 | 44.0% | 12.6 | 32.4% | 6.4 |
| | O | 10 | 50.0% | o | 1.5 | 32.3% | 21.4 | 55.3% | 4.7 |
| | P | 2 | 10.0% | p | 1.1 | 23.7% | 4.7 | 12.3% | 4.2 |
| | Total | 20 | 100.0% | total | 4.6 | 100.0% | 38.7 | 100.0% | 5.2 |

[1] "A": East Nanjing Road–The Bund; [2] "B": West Nanjing Road–Xinzha Road; [3] "C": Xujiahui–Xujiahui Garden; [4] "D": Jiangning Road; [5] "E": Yanggao Road; [6] "R": Stores on the ground floor of residential buildings; [7] "O": Stores in office buildings; [8] "P": Stores in public buildings; [9] "r": Residential space; [10] "o": office space; [11] "p": public space.

### 4.3. Relevance to Urban Spatial Functions

The five areas with the highest distribution density of convenience stores in Shanghai outer ring were selected, and the distribution locations of convenience stores were divided into three categories—commercial ground floor residential buildings, office buildings, and public buildings—as shown in Figure 8. The number of different locations of convenience stores in five high-density areas was counted, with urban space functions divided into three categories: living space, office space, and public space. The building area of each function and the ratio of each function were calculated. Statistics are shown in Table 3.

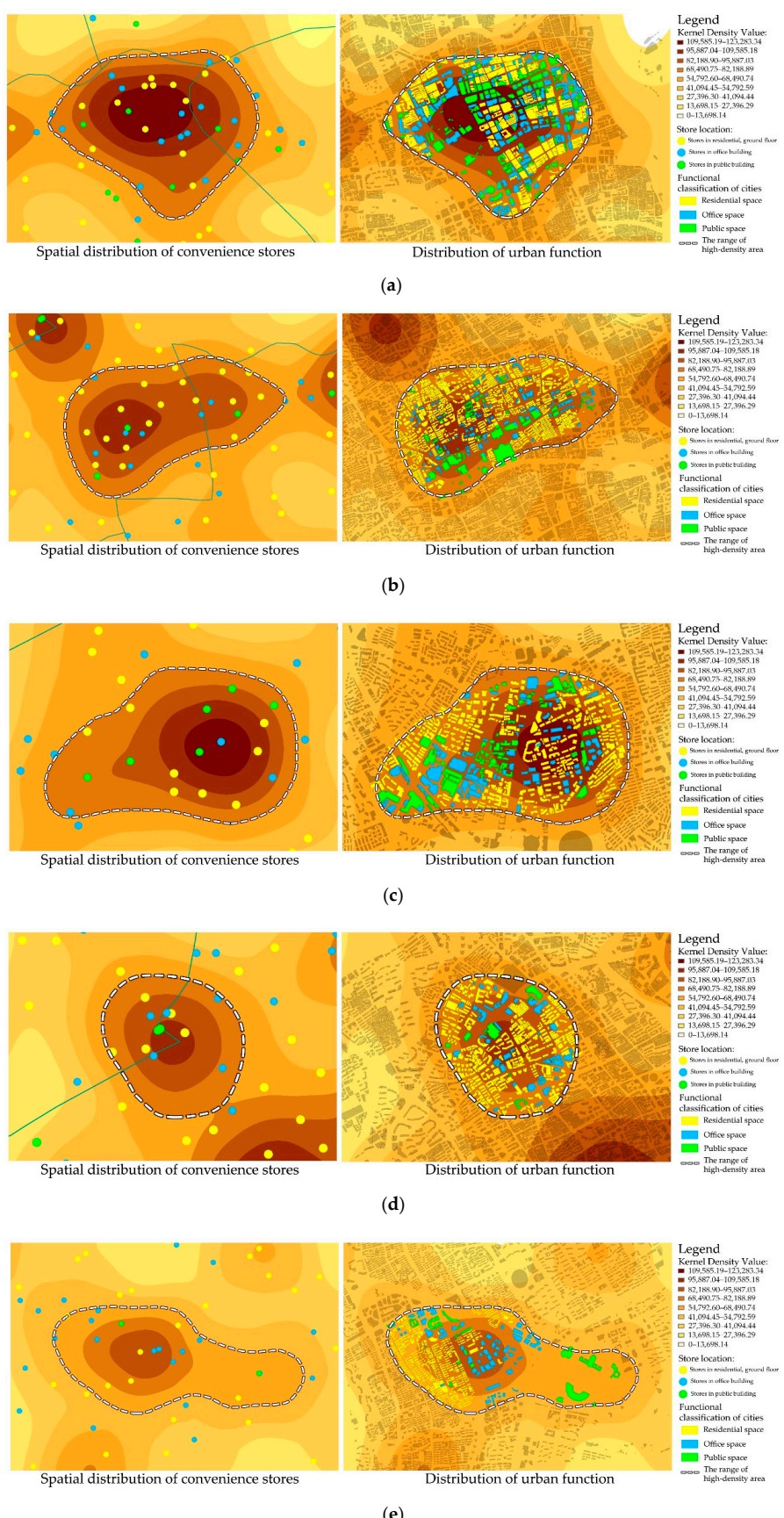

**Figure 8.** *Cont*.

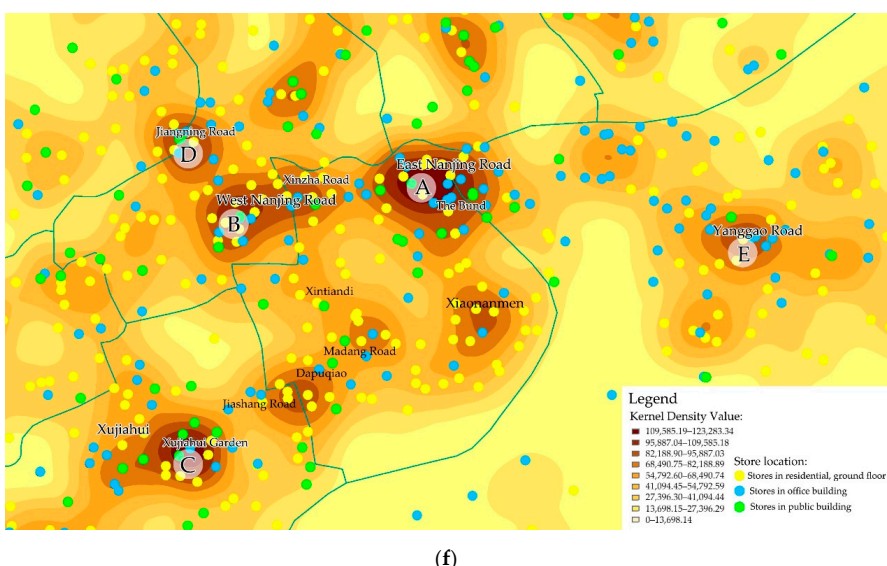

(**f**)

**Figure 8.** The superposition of POIs and the kernel density of convenience stores in different distribution locations. (**a**) High-density zone A. (**b**) High-density zone B. (**c**) High-density zone C. (**d**) High-density zone D. (**e**) High-density zone E. (**f**) Five high-density areas.

The proportions of the distribution locations of convenience stores in the high-density areas and the ratios of the functions of the urban space in the high-density areas were contrasted and analyzed. Figure 9 illustrates the result. In the four high-density areas A, B, C, and E, the distribution ratio of convenience store locations matched the distribution ratio of urban space functions, whereas the ratio of living space to office space in high-density area D was abnormal.

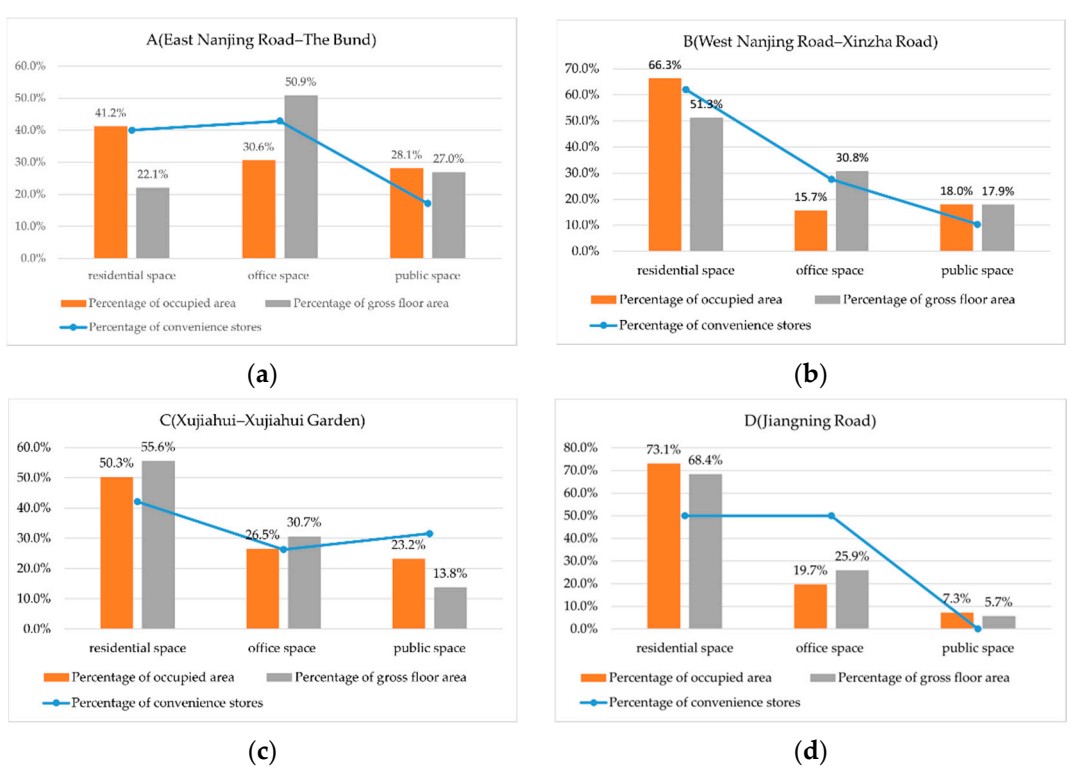

**Figure 9.** *Cont.*

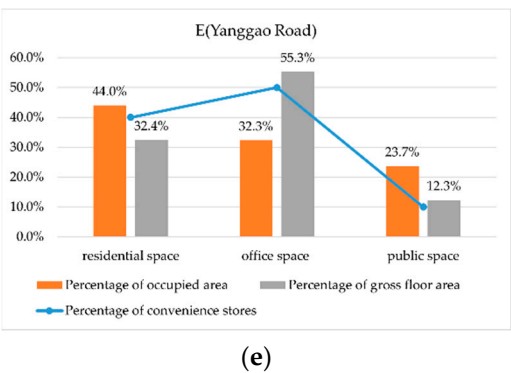

(**e**)

**Figure 9.** Correlation analyses between the spatial distribution of convenience stores and urban function distribution in the high-density areas. (**a**) Correlation analyses in high-density area A; (**b**) Correlation analyses in high-density area B; (**c**) Correlation analyses in high-density area C; (**d**) Correlation analyses in high-density area D; (**e**) Correlation analyses in high-density area E.

The number of convenience stores per square kilometer in the five areas with the highest convenience distribution density was counted, as shown in Table 3. In the three high-density areas of A, B, and C in the city center, the number of FamilyMart convenience stores per square kilometer of office space was between three and four. The number of FamilyMart convenience stores per square kilometer of residential area varied greatly, ranging from two to eight. The number of FamilyMart convenience stores per square kilometer in the five high-density areas was between three and five.

## 5. Discussion

The results show that in areas with a high distribution density of convenience stores, a distribution state that is highly coupled with urban spatial functions has been formed, which has the conditions to become a network foundation closely integrated with the community.

Firstly, the spatial distribution of FamilyMart convenience stores in the outer ring of Shanghai is centered on the three central urban areas of Huangpu District, Jing'an District, and Xuhui District.

Secondly, the high-density areas of convenience stores overlap with the main center of the city, i.e., the A (Nanjing East Road–The Bund) area, B (Nanjing West Road–Xinzha Road) area, C (Xujiahui–Xujiahui Garden) area, although there are also non-overlapping areas, such as D (Jiangning Road) and E (Yanggao High Road). Although FamilyMart originated in Japan, the location selection adhered to the strategy of initially opening stores which were mainly distributed in the vicinity of the main business district and in the places closest to the concentration of customers. However, in the development of convenience stores in China, they have gradually combined with China's national conditions, exhibiting different characteristics. The main center of the city integrates financial services, economic headquarters, business offices, culture and entertainment, innovation and creativity, tourism, and other functions, and the flow of people is high; therefore, a large number of convenience stores have been distributed to provide convenient services for office workers. Convenience stores in the D (Jiangning Road) area and E (Yanggao Gao Road) area, which are not in the main center of the city, are densely distributed. Through qualitative analysis, the reason was that the business structures around the D (Jiangning Road) area of Putuo District included subway stations, commercial office buildings, residential areas, Changshou Park, the M50 Creative Park, Suzhou River, etc. Century Park, the Shanghai Science and Technology Museum, residential areas, and subway stations were near the E (Yanggao Road) area in Pudong New Area, all around the "hot spots" of FamilyMart convenience stores. These facilities attract people, so the convenience stores are densely distributed.

Thirdly, in the five areas with the highest distribution density of convenience stores, the proportions of building functions where the storefronts are located roughly matches the proportion of urban space function distribution in the area. The number is relatively stable;

however, the number of FamilyMart convenience stores per square kilometer of living area varies greatly. However, there is also a mismatch between the ratio of building functions in the D (Jiangning Road) area and the distribution of urban spatial functions in this area. The reason behind the investigation is that the D (Jiangning Road) area is mainly surrounded by residential buildings, with a large number creative parks, such as the M50 Creative Park. The structures here are mainly low-rise buildings; thus, the living area accounts for a large proportion. The service groups of convenience stores are mainly residents and office workers; therefore, the distribution ratio of convenience stores to urban space functions is presented. As such, there is a distribution ratio mismatch.

Due to the need for the latest data to support this study and the limitations of data acquisition, the results of the study will be affected to a certain extent. At the same time, the aim of this research was to analyze the current situation of existing small retail businesses, and thus cannot be used to predict the future distribution trends of small retail business.

## 6. Conclusions

This paper mainly discusses the correlation between the spatial distribution characteristics of FamilyMart convenience stores in the outer ring of Shanghai and the distribution of urban spatial functions, which has been researched at the meso level. The research results show that convenience stores are highly coupled with residential and office communities in terms of macroscopic spatial distribution and have formed a network space foundation closely related to community life. At the same time, convenience stores exhibit characteristics of abundant traffic and high-frequency use. In future transformations to new retail commercial space, they have the capability to become spatial nodes that promote community exchanges and stimulate the sustainability of community vitality.

In addition to convenience stores, small retail businesses such as community coffee shops and bookstores may become nodes of community communication. With the rapid development of new retail in China, the combination of online and offline stores relying on Internet technology, small retail businesses in offices and residential communities can become carriers of community communication space. Spatial transformation design, such as reducing the number of store shelves, function replacement, shelf design, seat arrangement, etc., can be applied to provide more space for catering. Community gatherings, community exhibitions, community exchange meetings, and other activities can also be held to transform small retail businesses into public communication spaces with more complex functions, promote community exchanges, and stimulate the sustainable development of community vitality.

**Author Contributions:** Conceptualization, K.L.; Data curation, S.L., K.L. and Y.Y.; Formal analysis, S.L. and K.L.; Funding acquisition, K.L. and Y.L.; Investigation, S.L. and Y.Y.; Methodology, S.L. and K.L.; Project administration, K.L.; Software, S.L. and K.L.; Writing—original draft, S.L.; Writing—review and editing, K.L. and Y.L. All authors have read and agreed to the published version of the manuscript.

**Funding:** This study was sponsored by the Shanghai Planning Office of Philosophy and Social Science, Grant No. 2018ECK005 and No. 2018BCK002.

**Institutional Review Board Statement:** Not applicable.

**Informed Consent Statement:** Not applicable.

**Data Availability Statement:** The main sources of data in this study were the official website of FamilyMart, the geographic coordinate data of Baidu Map of FamilyMart in 2021, and the point of interest (POI) data of FamilyMart. The datasets used and/or analyzed during the current study are available from the corresponding author upon reasonable request.

**Conflicts of Interest:** The authors declare no conflict of interest.

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
