# Peer review of "The Correlation between Convenience Stores’ Distribution and Urban Spatial Function: Taking the FamilyMart Stores in Shanghai as an Example"

_sustainability, doi:10.3390/su14159457_

Round 1

Reviewer 1 Report

The paper analysis the correlation between the spatial distribution of convenience stores and the distribution of urban spatial functions. The topic is interesting, and the introduction, research method and results are clear and well written, but I am not sure the paper has suitable for publication in this journal. I didn’t find the relation with sustainability topics. Also, the literature view section is missing.

Author Response

What are the contributions of the paper:

The paper analysis the correlation between the spatial distribution of convenience stores and the distribution of urban spatial functions. The topic is interesting.

Thank you for all summary of the contributions for this paper.  

What are the additional ways in which the paper could be improved:

  1. I am not sure the paper has suitable for publication in this journal. I didn’t find the relation with sustainability topics

We have added relevant text to complement this study's relevance to sustainability in the Abstract and Introduction.

We believe that, in future, with the development of small retail businesses, these can be transformed into public activity spaces through space transformation and design, which promotes the sustainable development of community interaction and traditional retail business, stimulating the sustainability of community vitality.

Therefore, we believe that small retail commercial space has the possibility to become a community communication space. This can not only help the traditional retail and commercial space to effectively transform and achieve sustainable development, but also stimulate the sustainable vitality of the community public space.

  1. Also, the literature view section is missing.

We list the literature review separately as the second chapter.

Reviewer 2 Report

Review of the manuscript

 „ Study on the Correlation between Spatial Distribution Characteristics of  Convenience Stores and Urban Spatial Function Distribution − Take the  FamilyMart Convenience Store in Shanghai Outer Ring as an Example”

The paper is interesting and well-composed. The topic is interesting, but the article needs improvement in several aspects.

The abstract needs to include results. The title of the article is too long, needs to be shortened.

The novelty of the article should be highlighted in introduction section. Why authors have chosen the FamilyMart Convenience Store? The aim and motivation of the study should be presented more clearly.

The methodology section is presented too briefly, needs to be expand, especially “Qualitative and Quantitative approach” section.

Map legends need improvement (Figure 3, 4, 6). Large values of the data should be at the top of the legend, small values at the bottom.

The discussion section should be improved. It should include references to the literature from the introduction section. Please create a separate discussion section and show the disadvantages of the presented methods.

Please, carefully check the English language.

Reviewer 3 Report

The study is based on a methodologically detailed analysis and focuses on a well-defined case study. The study presents the development of the phenomenon at several scale levels, which is represented by high-level visualization.

At the same time, the questioning and conclusion of the study are not universal enough. It would be necessary to interpret the model-likeness of the narrower-scope case study in a broader sense.

In the introduction of the study, the question that goes beyond the specific example should also be clearly formulated, this can help in the conclusion to give conclusions that are more model-like and can be used in a wider interpretation space.

The conclusion of the study should be explained in more detail, in this form it is too rough and general, while it only applied to the case study. It would also be necessary to demonstrate the range of interpretation and adaptability of the model-like examination. This should also be interpreted and presented in connection with the literature.

Author Response

Reviewer Comments for Authors:

What are the contributions of the paper:

The study is based on a methodologically detailed analysis and focuses on a well-defined case study. The study presents the development of the phenomenon at several scale levels, which is represented by high-level visualization.

 Thank you for all summary of the contributions for this paper.

What are the additional ways in which the paper could be improved:

  1. At the same time, the questioning and conclusion of the study are not universal enough. It would be necessary to interpret the model-likeness of the narrower-scope case study in a broader sense.

We have made a more model-based summary in the conclusion section, and the research conclusions of this paper will also be applicable to more small-scale retail businesses in the community.

In addition to convenience stores, small retail businesses such as community coffee shops and bookstores may become nodes of community communication. With the rapid development of new retail in China, the combination of online and offline stores relying on Internet technology, small retail businesses in offices and residential communities can become carriers of community communication space. Spatial transformation design, such as reducing the number of store shelves, function replacement, shelf design, seat arrangement, etc., can be applied to provide more space for catering. Community gatherings, community exhibitions, community exchange meetings, and other activities can also be held to transform small retail businesses into public communication spaces with more complex functions, promote community exchanges, and stimulate the sustainable development of community vitality.

  1. In the introduction of the study, the question that goes beyond the specific example should also be clearly formulated, this can help in the conclusion to give conclusions that are more model-like and can be used in a wider interpretation space.

In the introduction, we supplemented the reasons for choosing FamilyMart, the purpose and motivation of the research, and stated that the conclusions can be used in more small retail businesses in the community, which is universal.

With the rapid rise of new retail in China, a new retail model, with a deep integration of online service, offline experience, and modern logistics, has emerged. More and more Internet companies have invested in the traditional supermarket industry and guided them to realize Internet transformation, such as Alibaba's Hema Xiansheng, JD.com's stake in Yonghui Supermarket, Amazon's acquisition of Whole Foods Market, etc. At the same time, under the background of the current online retail encountering rising bottlenecks, the difficult development of brick-and-mortar retail, and the emergence of urban communication crisis, traditional community retail is also exploring a direction for transformation. We found that in the current process of urban community space renewal, as the daily living space of residents, community public space still has problems, such as spatial simplification, excessive typification, and a low sense of belonging for residents. It is manifested in the lack of flow of people in the community public space and the low utilization rate, which cannot meet the diverse needs of community residents.

Through on-the-spot research, we found that small retail businesses in the community are gathering places for community residents. Whether they are in-office communities or residential communities, they all have the characteristics of high traffic and high frequency of use. Residents have strong communication needs in small community retail businesses. For example, community coffee shops, bookstores, and retail stores will attract large numbers of people. People stop here to rest and chat. Therefore, we believe that small retail commercial space has the possibility to become a community communication space. This can not only help the traditional retail and commercial space to effectively transform and achieve sustainable development, but also stimulate the sustainable vitality of the community public space. At present, there are few existing studies on the combination of community retail commercial space and community interaction space. Therefore, this research has a certain timeliness and pertinence.

We have selected a representative small retail business, FamilyMart, which is the oldest small retail business with the largest volume and widest distribution in Shanghai. First, we discuss its overall spatial distribution characteristics. Additionally, we further compare its spatial distribution with the urban spatial function distribution to verify whether the two have formed a strong coupling relationship. Conditioned to provide a space foundation for the occurrence of community public communication activities, the organization can undertake more community communication functions in the future and realize the sustainable development of traditional retail commercial space.

  1. The conclusion of the study should be explained in more detail, in this form it is too rough and general, while it only applied to the case study. It would also be necessary to demonstrate the range of interpretation and adaptability of the model-like examination. This should also be interpreted and presented in connection with the literature.

We explain the findings in more detail and supplement the generality of the findings.

This paper mainly discusses the correlation between the spatial distribution characteristics of FamilyMart convenience stores in the outer ring of Shanghai and the distribution of urban spatial functions, which has been researched at the meso-level. The research results show that convenience stores are highly coupled with residential and office communities in terms of macroscopic spatial distribution and have formed a network space foundation closely related to community life. At the same time, convenience stores exhibit characteristics of abundant traffic and high-frequency use. In future transformations to new retail commercial space, they have the capability to be-come spatial nodes that promote community exchanges and stimulate the sustainability of community vitality.

In addition to convenience stores, small retail businesses such as community coffee shops and bookstores may become nodes of community communication. With the rapid development of new retail in China, the combination of online and offline stores relying on Internet technology, small retail businesses in offices and residential communities can become carriers of community communication space. Spatial transformation design, such as reducing the number of store shelves, function replacement, shelf design, seat arrangement, etc., can be applied to provide more space for catering. Community gatherings, community exhibitions, community exchange meetings, and other activities can also be held to transform small retail businesses into public communication spaces with more complex functions, promote community exchanges, and stimulate the sustainable development of community vitality.

Round 2

Reviewer 1 Report

The manuscript has been improved sufficiently.

Reviewer 2 Report

In general, the authors try to insert the suggestions that were made and answer to all the questions in a satisfying way. I suggest to published article in present form.

Congratulations for you work!